# A Weight Variation-Aware Training Method for Hardware Neuromorphic Chips

## Abstract

Hardware neuromorphic chips that mimic the biological nervous systems have recently attracted significant attention due to their ultra-low power and parallel computation. However, the inherent variability of nano-scale synaptic devices causes a weight perturbation and performance drop of neural networks. This paper proposes a training method to find weight with robustness to intrinsic device variability. A stochastic weight characteristic incurred by device inherent variability is considered during training. We investigate the impact of weight variation on both Spiking Neural Network (SNN) and standard Artificial Neural Network (ANN) with different architectures including fully connected, convolutional neural network (CNN), VGG, and ResNet on MNIST, CIFAR-10, and CIFAR-100. Experimental results show that a *weight variation-aware training method (WVAT)* can dramatically minimize the performance drop on weight variability by exploring a flat loss landscape. When there are weight perturbations, WVAT yields 85.21% accuracy of VGG-5 on CIFAR-10, reducing accuracy degradation by more than 1/10 compared with SGD. Finally, WVAT is easy to implement on various architectures with little computational overhead.

## 1 Introduction

Deep Neural Networks (DNN) have achieved remarkable breakthroughs in computer vision, automatic driving, and image/voice recognition (LeCun et al., 2015). With this success, neuromorphic technology, which mimics the human nervous system, has recently received significant attention in the semiconductor industry. Compared with the conventional von Neumann architecture which has limitations in power consumption and real-time pattern recognition (Schuman et al., 2017; Indiveri et al., 2015), neuromorphic chips, biologically inspired from the human brain, are new compact semiconductor chips that collocate processing and memory (Chicca et al., 2014; Catherine D. Schuman & Kay, 2022). Therefore, neuromorphic chips can process highly parallel operations and be suitable for real-time recognizing images, videos, and audios with ultra-low power consumption (Indiveri & Liu, 2015).

Neuromorphic chips are also suitable for "Edge AI computing," which process data in edge devices rather than in the cloud at a data center (Nwakanma et al., 2021). In other words, tasks that require a large amount of computation, such as training, are performed in the cloud and inference in edge devices. Traditional cloud AI processing requires sufficient computing power and network connectivity. This means that an enormous amount of data transmission is required, likely increasing data latency and transferring disconnections (Li et al., 2020). It causes severe problems in autonomous driving, robotics, and mobile VR/AR that require real-time processing. Therefore, there is a growing need for data processing on edge devices. Neuromorphic devices are compact, mobile, and energy-efficient, promising candidates for edge computing systems.

However, despite enormous advances in semiconductor integrated circuit (IC) technology, hardware neuromorphic implementation and embedded systems with numerous synaptic devices remain challenging (Prezioso et al., 2015; Esser et al., 2015; Catherine D. Schuman & Kay, 2022). Design considerations such as multi-level state, device variability, programming energy, speed, and array-level connectivity, are required. (Eryilmaz et al., 2015). In particular, nano-electronic device variability is an inevitable issue originating from manufacturing fabrication (Prezioso et al., 2010). Although there are many kinds of nano-electronic devices for neuromorphic systems and in-memory

computing–including memristor, flash memory, phase-change memory, and optoelectronic devices–we call them "devices" for readability in this paper.

Device variability causes mapped synaptic weight values in hardware to differ undesirably from software weight. This gap between hardware synapse and software weight makes it challenging to implement neural networks in real-world applications. Many recent studies have reported that device variability can significantly reduce the accuracy of neuromorphic hardware and DNN accelerators (Catherine D. Schuman & Kay, 2022; Peng et al., 2020; Joshi et al., 2020b; Sun & Yu, 2019; Kim et al., 2019; 2018). Although there are various studies to solve this problem, they focus on the unique behaviors of devices (Hennen et al., 2022; vls; Fu et al., 2022). The diversity of devices used to implement neuromorphic hardware results in the customized solutions required for a given device variation. Therefore, the versatility of customized solutions at the device level is limited.

There is a growing need for a hardware-oriented training method to learn parameters robust to device variability. It is widely known that *wide and flat loss landscapes* lead to improved generalization (Keskar et al., 2017; Li et al., 2018). It is natural to think that wide and flat loss landscapes with respect to weight will mitigate the accuracy drop caused by device variability. However, we experimentally confirm that related studies (Izmailov et al., 2018; Wu et al., 2020; Foret et al., 2021) can not significantly reduce the accuracy drop by device variation (Experiments are provided later in section 2). This observation reminds us of the need for a hardware-oriented neural network training method.

Motivated by this, we propose a *weight variation-aware training method (WVAT)* that alleviates performance drops induced by device variability at the algorithmic level rather than the device level. This method explores a wide and flat *weight loss landscape* through the ensemble technique and the hardware-simulated variation-aware update method, which is more tolerant to the software weight perturbation caused by hardware synaptic variability. WVAT can effectively minimize performance drops with respect to weight variations with little additional computational overhead in the training phase. Our contributions include the following:

- For the first time to the best of our knowledge, we investigate and analyze the impact of variations in model parameters on performance in several architectures, including standard Artificial Neural Networks (ANN) and Spiking Neural Networks (SNN), which are suitable for hardware neuromorphic implementation due to event-driven spike properties.
- By exploring the flatter weight loss landscape, we propose WVAT that is tolerant to intrinsic device variability. We introduce an ensemble technique for better generalization and present a intuitive weight update method with a hardware-simulated variation. This method is also efficient for quantization and input noise, which is one of the hardware implementation issues besides weight perturbations.
- We experimentally demonstrate that WVAT achieves nearly similar performance to the typical training method stochastic gradient descent (SGD) while having robustness to variations in model parameters. WVAT is easy to implement with little computational cost.
- By presenting an algorithm-level hardware-oriented training method, we expect that WVAT will help the development of society related to hardware implementation, including neuromorphic chips[1].

## 2 BACKGROUND

Many studies have been conducted to develop training methods robust to device variability (Liu et al., 2015; Long et al., 2019; Zhu et al., 2020; Joshi et al., 2020b; Joksas et al., 2022; Huang et al., 2022). Liu et al. (2015) proposed adding a penalty for variations in model parameters to training loss, and Long et al. (2019); Zhu et al. (2020) generated a noise model to reflect device variability during a training phase. Although Long et al. (2019) achieved good performance, this method has a limitation, a binary device (1 bit per cell). However, as mentioned in section 1, the customized solutions for the given device has limitation in applying to the general case.

Recently, there have been many studies investigating the effect of loss landscape on generalization (Garipov et al., 2018; Izmailov et al., 2018; Wu et al., 2020; Foret et al., 2021; Liu et al., 2022). It is

---

[1] A source code will be available soon.

widely known that a flat loss landscape reduces the generalization gap. Stochastic weight averaging (SWA) (Izmailov et al., 2018) is an ensemble technique that averages the weights as a time axis instead of storing multiple models with different weights. It has been experimentally demonstrated that the ensemble method brings a flatter loss landscape, resulting in a lower test error than SGD. Adversarial weight perturbation (AWP) (Wu et al., 2020) and sharpness-aware minimization (SAM) (Foret et al., 2021) were proposed to improve model generalization by seeking the flat loss landscape. These methods explore the direction of weight perturbation in the worst case and update the weight based on that direction.

**AWP** AWP proposed a adversarial weight perturbation $\boldsymbol{v}$ based on the generated adversarial examples $x'_i$ for adversarial training:

$$\boldsymbol{v} = \frac{\nabla_{\boldsymbol{v}} \frac{1}{m} \sum_{i=1}^{m} l(f_{\boldsymbol{w}+\boldsymbol{v}}(x'_i), y_i)}{||\nabla_{\boldsymbol{v}} \frac{1}{m} \sum_{i=1}^{m} l(f_{\boldsymbol{w}+\boldsymbol{v}}(x'_i), y_i)||} ||\boldsymbol{w}||$$

where $y_i$ is label, and $m$ is batch size. $f_{\boldsymbol{w}}(\cdot)$ is neural network with weight $\boldsymbol{w}$, and $l$ is standard classification loss.

**SAM** SAM seeks model parameters whose entire neighborhoods have uniformly low training loss value. Neighborhood $\epsilon(w)$—maximizing loss value— is given by the solution of a classical dual norm problem:

$$\epsilon(\boldsymbol{w}) = \rho \, \text{sign}(\nabla_{\boldsymbol{w}} L_S(\boldsymbol{w})) |\nabla_{\boldsymbol{w}} L_S(\boldsymbol{w})|^{q-1} / (||\nabla_{\boldsymbol{w}} L_S(\boldsymbol{w})||_q^q)^{\frac{1}{p}}$$

where $1/p + 1/q = 1$, $\rho$ is neighborhood size, and $L_S(\cdot)$ is training set loss. In the case of $p = 2$, $\epsilon(\boldsymbol{w})$ is a norm of the gradient scaled by $\rho$.

Both methods have similarities in finding weight perturbations in the gradient-ascent direction, except the perturbation is scaled by a norm of weight (AWP) or neighborhood size (SAM). Although both methods yielded state-of-the-art performance, they have drawbacks in terms of computational overhead . A update rule in AWP and SAM requires two sequential gradient computations, one for obtaining weight perturbation and another for computing the gradient descent update (Liu et al., 2022). This has twice the computational overhead compared with SGD.

The effect of weight perturbation induced by device variability on model performance can also be considered a kind of generalization problem. We start with a perspective that the flat loss landscape will have robustness against variations in model parameters. A naive approach to solve device variability issues is applying techniques related to generalization. Unfortunately, as shown in Figure 1, these techniques did not lead to a significant improvement in weight variation. As mentioned above, since AWP and SAM are similar algorithms in terms of computing weight perturbations, the experiment was conducted with AWP. When there are variations in the weight, an accuracy drop of 36.58% for SGD, 34.17%

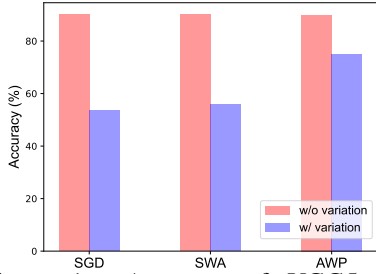

Figure 1: Accuracy of VGG5 on CIFAR-10 using SGD, SWA, and AWP.

for SWA, and 14.89% for AWP occurs compared with a case without variation. The accuracy drop caused by device variability is one of the major issues that make hardware implementation a challenge. This motivates us to explore a new hardware-oriented neural network training method. Therefore, we proposed efficient and intuitive WVAT by considering stochastic weight characteristics incurred by device inherent variability in the training phase and in the weight update to flatten the loss landscape.

## 3 WEIGHT VARIATION-AWARE TRAINING (WVAT)

Our goal is to find parameters that are tolerant to perturbations of model parameters (e.g., weights) caused by the variability of synaptic hardware devices. We aim to minimize the loss, along with the difference in the loss with respect to perturbed weights. Thus, objective is as follows:

$$\min_{\boldsymbol{w}} [L(\boldsymbol{w}) + (L(\boldsymbol{w} + \boldsymbol{w_v}) - L(\boldsymbol{w}))] \rightarrow \min_{\boldsymbol{w}} L(\boldsymbol{w} + \boldsymbol{w_v})$$

where where $\boldsymbol{w}$ and $\boldsymbol{w_v}$ denotes weight and weight variation, respectively. $L(\cdot)$ represents loss function. In order to explore the flat loss landscapes with respect to weight variation, the difference term $L(\boldsymbol{w} + \boldsymbol{w_v}) - L(\boldsymbol{w})$ should be minimized. When the weight trained in software (cloud) is transferred to a hardware device (edge device), the mapped weight is likely to be different from the software weight due to device variability. This results in performance degradation. For these reasons, here we devise two types of variations that reproduce device variations during a training phase. One is a *hardware-simulated variation (HSV)* reflecting device variability, and the other is a *gradient-ascent variation (GAV)*.

**Hardware-simulated variation (HSV)**

Assume "analog" synaptic devices, as shown in Figure 2, the layer-wise weight range $A$ is set $[\mu - 3\sigma, \ \mu + 3\sigma]$ across the weight distribution, and we take a weight variation on $A$.

$$\Delta \boldsymbol{w} = \gamma \mathcal{N}(0, \ A\sigma_v{}^2)$$

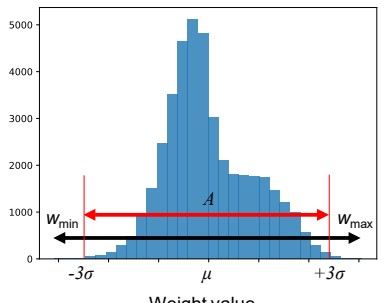

where $\Delta \boldsymbol{w}$ is a *hardware-simulated variation (HSV)* to imitate intrinsic hardware device variability, which is a random sample of the same size as $\boldsymbol{w}$ from a Gaussian distribution with mean 0 and variation $A\sigma_v$. $\gamma$ is a range coefficient, scaling factor, to determine the variation size. In general, when mapping software weights to a synaptic hardware device, clipping method is widely used rather than the full range of software weights ($[w_{min}, w_{max}]$) due to the memory window limit of the synaptic device (Kwon et al., 2019; Joshi et al., 2020b). Therefore, $A$ is

Figure 2: Weight distribution of a layer. A graph indicates the histogram of the software weight. Due to the device variability, the original software weight is perturbed. HSV is drawn from the weight range $A$ to mimic the variation.

the range of weights that can be expressed in synaptic hardware devices. HSV refers to how much variation occurs within the range that can be expressed in the device.

Many studies have reported that fabricated synaptic devices have Gaussian distribution (Gong et al., 2018; Boybat et al., 2018; Yu et al., 2013); therefore, we use a Gaussian distribution when generating HSV. In addition, considering that the standard deviation of the fabricated devices is usually $\sim5\%$ (Joshi et al., 2020a; Wan et al., 2019), we set that a 10% standard deviation was simulated during training for more stable results. For example, $\sigma_v$ of 10% means that the weight has changed by 10% of $A$. In order to minimize objective $L(\boldsymbol{w} + \Delta \boldsymbol{w})$ using SGD as a optimizer, differentiating it as follows:

$$\nabla_{\boldsymbol{w}} L(\boldsymbol{w} + \Delta \boldsymbol{w}) = \frac{d(\boldsymbol{w} + \Delta \boldsymbol{w})}{d\boldsymbol{w}} \frac{dL(\boldsymbol{w})}{d\boldsymbol{w}} \bigg|_{\boldsymbol{w} = \boldsymbol{w} + \Delta \boldsymbol{w}}$$

$$= 1 \nabla_{\boldsymbol{w}} L(\boldsymbol{w})|_{\boldsymbol{w} = \boldsymbol{w} + \Delta \boldsymbol{w}}$$

$\nabla_{\boldsymbol{w}} L(\boldsymbol{w} + \Delta \boldsymbol{w})$ can be calculated as the differentiation at the value in which the weight variation occurs by the differentiation of the composite function.

**Gradient-ascent variation (GAV)** In addition to reproducing HSV during the training phase, the weight variation corresponding to the worst-case—making the greatest the difference term—should also be reflected to find flat loss landscapes. Recalling our objective, this is modified as a maximization problem.

$$\min_{\boldsymbol{w}}[L(\boldsymbol{w}) + \max_{\boldsymbol{v}(\boldsymbol{w})}(L(\boldsymbol{w} + \boldsymbol{w_v}) - L(\boldsymbol{w}))] \rightarrow \min_{\boldsymbol{w}} \max_{\boldsymbol{w_v}} L(\boldsymbol{w} + \boldsymbol{w_v})$$

$L(\boldsymbol{w} + \boldsymbol{w_v})$ can be approximated by first-order Taylor expansion to find the weight variation $\boldsymbol{w_v}$ that maximizes the loss.

$$L(\boldsymbol{w} + \boldsymbol{w_v}) = L(\boldsymbol{w} + \alpha \boldsymbol{v}(\boldsymbol{w})) \approx L(\boldsymbol{w}) + \alpha \boldsymbol{v}(\boldsymbol{w})^T \nabla_{\boldsymbol{w}} L(\boldsymbol{w})$$

where $\alpha$ is a step-length parameter. $\boldsymbol{v}(\boldsymbol{w})^T \nabla_{\boldsymbol{w}} L(\boldsymbol{w})$ is the rate of change in $L$ along the direction $\boldsymbol{v}(\boldsymbol{w})$ at $\boldsymbol{w}$. Therefore, the most rapidly increasing direction is the solution to the problem.

$$\max_{\boldsymbol{v}(\boldsymbol{w})} \boldsymbol{v}(\boldsymbol{w})^T \nabla_{\boldsymbol{w}} L(\boldsymbol{w}) \qquad \text{subject to } ||\boldsymbol{v}(\boldsymbol{w})|| = ||\Delta \boldsymbol{w}||$$

$$\boldsymbol{v}(\boldsymbol{w}) = \frac{\nabla L_{\boldsymbol{w}}(\boldsymbol{w})}{||\nabla L_{\boldsymbol{w}}(\boldsymbol{w})||} ||\Delta \boldsymbol{w}||$$

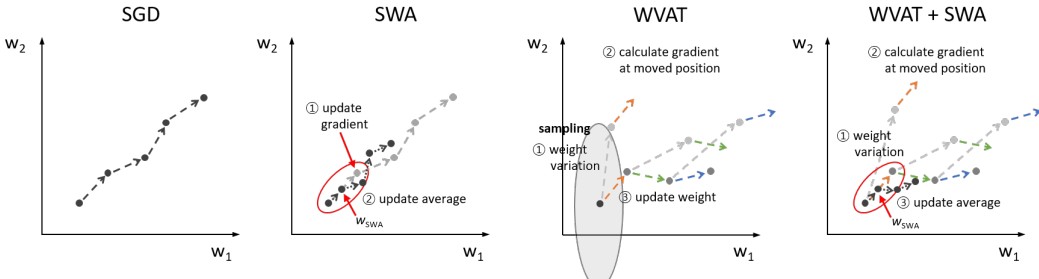

Figure 3: Schematic of the weight update of the each method.

where the magnitude of the weight variation is set to be the same as that of HSV. $\nabla L_{\boldsymbol{w}}(\boldsymbol{w})$ is the steepest ascent direction for a line search method (Nocedal & Wright, 1999). For this reason, we name $\boldsymbol{v}(\boldsymbol{w})$ a *gradient-ascent variation (GAV)*. This is a similar approach to AWP and SAM.

**Weight update** Weight variation is simulated during training and updates the weight via SGD. The training process is as follows:

$$\text{Weight variation: } \boldsymbol{w} \leftarrow \boldsymbol{w} + \boldsymbol{w_v}$$
$$\text{Weight update: } \boldsymbol{w} \leftarrow \boldsymbol{w} - \eta \nabla L_{\boldsymbol{w}}(\boldsymbol{w} + \boldsymbol{w_v})$$
$$\text{Weight reverse: } \boldsymbol{w} \leftarrow \boldsymbol{w} - \boldsymbol{w_v}$$

where $\eta$ is a learning rate. In this case, $w_v$ can be one of the two proposed HSV and GAV. Depending on the probability $p$, it determine whether or not to adding $w_v$ for each batch. If no variance is added, the weight update is equivalent to SGD and no weight reverse is performed.

$$x \sim U(0, 1)$$
$$\text{HSV } \Delta \boldsymbol{w} \text{ is generated if } x < w_{th}$$
$$\text{GAV } \boldsymbol{v}(\boldsymbol{w}) \text{ is generated if } x \geq w_{th}$$

$w_{th}$ is a threshold of what kind of weight variation it will reproduce during training. $x$ is a randomly sampled value from a uniform distribution for each batch. For each batch, $x$ determines what kind of variation will be generated. Figure 3 schematically illustrates the weight update according to each method. SWA averages multiple points along the trajectory of SGD, leading to better generalization than SGD. By applying SWA to WVAT, wider loss landscapes can be explored.

## 4 EXPERIMENTS

In this section, we conduct experiments to evaluate the proposed WVAT on both artificial neural network (ANN) and spiking neural network (SNN) with different architectures, including fully connected (FC), convolutional neural network (CNN), VGG, and ResNet on benchmark datasets (MNIST, CIFAR-10, CIFAR-100). These benchmarks and models have been widely used in hardware implementations (Kim et al., 2018; Long et al., 2019; Zhu et al., 2020; Joshi et al., 2020b; Joksas et al., 2022; Huang et al., 2022; Jung et al., 2022). Edge devices are mainly implemented using small models; moreover, small models are vulnerable to performance degradation due to device variability. Hence, we focus on experiments on small models, including ablation studies and comparisons with SGD, temporal spike sequence learning via backpropagation (TSSL-BP) (Zhang & Li, 2020), and SWA. Although SWA yields better performance via longer training, all method is trained for the same epochs for a fair comparison. We report the mean and standard deviation of test accuracy over 5 runs.

Figure 4 (a) and (b) shows the effect of two hyperparmeters on model performance for VGG5 on CIFAR-10 and VGG16 on CIFAR-100, respectively. We test WVAT for hyperparameter $w_{th} \in \{0, 0.1, 0.2, 0.3, 0.4, 0.5, 0.6, 0.7, 0.8, 0.9, 1\}$. If $w_{th} = 0$, only HSV is reflected, and if $w_{th} = 1$, only GAV is generated during training. For a baseline, we test SGD and SGD Test Time Variation (SGD-TTV), which means when the weight of the trained model perturbs during a test phase. As shown in Figure 4 (a), SGD achieved 90.09% accuracy, while SGD-TTV 53.51% when $\sigma_v$ of 10% (36.58% performance drop by the variation). In the case of $w_{th} = 1$, only GAV is reproduced

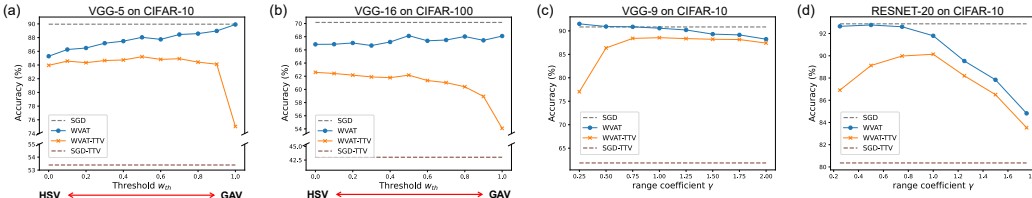

Figure 4: Accuracy of SGD and WVAT. (**a**), (**b**) Effect of HSV and GAV, and (**c**), (**d**) range coefficient $\gamma$ on model performance.

during the training phase. Although it achieved the highest accuracy, it did not effectively prevent the degradation in accuracy when there was a variation in the weight. On the other hand, in the case of $w_{th} = 0$, WVAT minimized the degradation in accuracy during TTV, but the model performance was poor than SGD. Therefore, we set $w_{th} = 0.5$ as the default value, which achieves nearly the similar performance as SGD and is robust to weight perturbation induced by device variability. Since HSV requires one gradient computation, like SGD, and GAV requires two sequential gradient computations, our proposed WVAT, which is a mixture of HSV and GAV, has less computational overhead compared to AWP and SAM. Through this experiment, we experimentally confirmed that reflecting both HSV and GAV is a key factor in model performance and robustness against stochastic weight characteristics.

Range coefficient $\gamma$ is a hyperparameter that determines the variation size simulated in training. $\gamma = 1$ means that the variation size during TTV is equally reflected during training. As shown in Figure 4 (c) and (d), HSV with the same variation size as the test minimizes the accuracy drop. We set $\gamma = 1$ as the default value. Engineers can choose $w_{th}$ and $\gamma$ by considering the trade-offs for accuracy when variations occur and when they do not, according to the characteristics of each device.

### 4.1 MNIST

For MNIST, we compare the accuracy of the ANN with the FC network and spiking CNN trained by each method in Table 1. FC network are trained using SGD optimizer with learning rate 0.0005. When applying SWA, we use SWA training with a fixed learning rate schedule from scratch. We train spiking-CNNs by TSSL-BP for training SNN, and all models for 100 epochs. For TSSL-BP, we use Adam optimizer with a learning rate of 0.0005 and other settings are the same as in Zhang & Li (2020). Hyperparameters for WVAT are set as default value, $\gamma = 1$, $w_{th} = 0.5$, and $\alpha = 0.1$. For each network architecture, the proposed WVAT achieves nearly similar performance as SGD and SWA and minimizes accuracy degradation when weight variation occurs.

Hardware devices inevitably have inherent variability. Therefore, in most cases, the model performance in software cannot be directly reproduced in hardware implementation. From the hardware implementation point of view, it is preferable to refer to the accuracy when there is weight variation rather than the accuracy when there is no variation. In the case of spiking-CNN1, when $\sigma_v = 10\%$, WVAT yields 96.02% accuracy achieving $\approx 1/5$ reduction of accuracy drop compared with TSSL-BP (16.97% performance drop for TSSL-BP, 3.29% for WVAT).

### 4.2 CIFAR-10

We test the proposed WVAT on CIFAR-10 in Table 2. For comparison experiments, we use VGG-5, VGG-16, ResNet-20, and ResNet-110 models. VGG-5, VGG-16 and ResNet-20 are trained for 200 epochs using SGD with momentum 0.9, weight decay 0.0005, and an initial learning rate of 0.01. We use Preactivation ResNet-110 in Garipov et al. (2018). ResNet-110 are trained for 150 epochs using SGD with momentum 0.9, weight decay 0.0003, and an initial learning rate of 0.1. When applying SWA, we first run SGD optimizer with a decaying learning rate schedule for 75% of the training budget, and then apply SWA with a fixed learning rate of 0.005 for all models except for 0.01 for ResNet-110. We adopt a hybrid conversion method (Rathi et al., 2020) for SNN implementation. Hyperparameters for WVAT are set as $\gamma = 1$, $w_{th} = 0.5$, and $\alpha = 0.1$ for VGG-5 and 0.01 for VGG-16, ResNet-20, and ResNet-110.

Table 1: Accuracy on MNIST for different model and method. 15C5 means convolution layer with 15 of the $5 \times 5$ filters, and P2 pooling layer with $2 \times 2$ filters.

| Model | Method | Acc (%) | Acc (%) $\sigma_v = 5\%$ | Acc (%) $\sigma_v = 10\%$ |
|---|---|---|---|---|
| FC 2-layer[1] (ANN) | SGD | $98.13 \pm 0.09$ | $97.86 \pm 0.14$ | $96.54 \pm 0.23$ |
| | SWA | $\mathbf{98.56} \pm 0.12$ | $98.31 \pm 0.71$ | $97.20 \pm 1.54$ |
| | WVAT | $98.47 \pm 0.05$ | $\mathbf{98.39} \pm 0.04$ | $\mathbf{97.98} \pm 0.19$ |
| Spiking-CNN1[2] (SNN) | TSSL-BP | $\mathbf{99.37} \pm 0.02$ | $98.08 \pm 0.15$ | $82.40 \pm 8.03$ |
| | SWA | $99.29 \pm 0.11$ | $98.62 \pm 0.60$ | $90.24 \pm 2.34$ |
| | WVAT | $99.31 \pm 0.07$ | $\mathbf{99.07} \pm 0.07$ | $\mathbf{96.02} \pm 0.82$ |
| Spiking-CNN2[3] (SNN) | TSSL-BP | $99.38 \pm 0.05$ | $98.81 \pm 0.06$ | $86.17 \pm 8.55$ |
| | SWA | $\mathbf{99.44} \pm 0.03$ | $98.23 \pm 0.08$ | $89.73 \pm 2.69$ |
| | WVAT | $99.39 \pm 0.03$ | $\mathbf{99.24} \pm 0.08$ | $\mathbf{95.57} \pm 1.87$ |

[1] 784-400-200-10.
[2] 16C5-P2-Concat(32C3, 8C1)-8C1-288.
[3] 15C5-P2-40C5-P2-300.

Table 2: Accuracy on CIFAR-10 for different model and method.

| Model | Method | Acc (%) | Acc (%) $\sigma_v = 5\%$ | Acc (%) $\sigma_v = 10\%$ |
|---|---|---|---|---|
| VGG-5 (ANN) | SGD | $89.96 \pm 0.16$ | $79.08 \pm 0.95$ | $53.39 \pm 3.92$ |
| | SWA | $\mathbf{89.97} \pm 0.22$ | $80.62 \pm 1.51$ | $57.14 \pm 2.42$ |
| | WVAT | $87.76 \pm 0.24$ | $\mathbf{87.05} \pm 0.25$ | $\mathbf{85.21} \pm 0.19$ |
| VGG-16 (SNN conversion) | SGD | $92.68 \pm 0.11$ | $90.49 \pm 0.08$ | $79.71 \pm 1.27$ |
| | SWA | $\mathbf{92.77} \pm 0.08$ | $91.05 \pm 0.16$ | $84.10 \pm 0.75$ |
| | WVAT | $91.79 \pm 0.144$ | $\mathbf{91.20} \pm 0.12$ | $\mathbf{89.25} \pm 0.10$ |
| ResNet-20 (ANN) | SGD | $92.88 \pm 0.23$ | $90.57 \pm 0.28$ | $80.36 \pm 1.17$ |
| | SWA | $\mathbf{92.98} \pm 0.19$ | $\mathbf{91.51} \pm 0.18$ | $84.54 \pm 0.57$ |
| | WVAT | $91.88 \pm 0.11$ | $91.43 \pm 0.12$ | $\mathbf{90.06} \pm 0.12$ |
| ResNet-110 (ANN) | SGD | $95.06 \pm 0.17$ | $93.58 \pm 0.12$ | $84.68 \pm 0.71$ |
| | SWA | $\mathbf{95.47} \pm 0.06$ | $\mathbf{94.26} \pm 0.09$ | $87.26 \pm 0.95$ |
| | WVAT | $94.45 \pm 0.16$ | $93.64 \pm 0.13$ | $\mathbf{90.47} \pm 0.12$ |

We experimentally confirm that WVAT minimizes the decrease in accuracy when there is variation induced by device variability in most models. Compared to MNIST, the performance degradation due to weight perturbation is more pronounced. In the case of VGG-5, when $\sigma_v = 10\%$, WVAT yields 85.21% accuracy, while 53.39% for SGD and 57.14% for SWA. WVAT reduces the performance drop by more than 1/10 compared with SGD and SWA. Therefore, we demonstrated that the advantage of WVAT in small models is more significant.

## 4.3 CIFAR-100

We compare the accuracy of each method on CIFAR-100, and the results are summarized in Table 3. For comparison experiments, we use VGG-16, ResNet-34, and ResNet-110 models. It is basically the same as the experimental setting of CIFAR-10, and the different parts is as follows. Experimental settings for ResNet-34 on CIFAT-100 are the same as ResNet-20 on CIFAR-10. ResNet-110 on CIFAR-100 is trained with an initial learning rate of 0.05. As mentioned in section 4.2, WVAT outperforms other methods not only in the small model but also in Resnet-110. It is important

to note that the standard deviation of accuracy using WVAT is also the smallest for each network architecture, which means that the proposed WVAT is tolerant to weight variation.

Table 3: Accuracy on CIFAR-100 for different model and method.

| Model | Method | Acc (%) | Acc (%) $\sigma_v = 5\%$ | Acc (%) $\sigma_v = 10\%$ |
|---|---|---|---|---|
| VGG-16 (ANN) | SGD | $70.17 \pm 0.19$ | $63.88 \pm 0.41$ | $43.02 \pm 0.76$ |
| | SWA | $\textbf{70.46} \pm 0.32$ | $66.33 \pm 0.25$ | $51.87 \pm 1.25$ |
| | WVAT | $68.13 \pm 0.12$ | $\textbf{66.51} \pm 0.14$ | $\textbf{62.16} \pm 0.11$ |
| ResNet-34 (ANN) | SGD | $61.40 \pm 0.68$ | $58.73 \pm 0.63$ | $49.80 \pm 1.04$ |
| | SWA | $61.28 \pm 1.46$ | $59.57 \pm 1.38$ | $53.33 \pm 1.33$ |
| | WVAT | $\textbf{63.65} \pm 0.43$ | $\textbf{62.49} \pm 0.40$ | $\textbf{59.25} \pm 0.65$ |
| ResNet-110 (ANN) | SGD | $76.72 \pm 0.55$ | $69.22 \pm 0.42$ | $32.68 \pm 0.75$ |
| | SWA | $\textbf{78.52} \pm 0.19$ | $74.33 \pm 0.14$ | $52.53 \pm 0.32$ |
| | WVAT | $78.13 \pm 0.94$ | $\textbf{74.59} \pm 0.87$ | $\textbf{54.25} \pm 0.82$ |

## 5 DISCUSSION

### 5.1 WEIGHT LOSS LANDSCAPES

In this section, an experiment is first conducted to investigate the effect of the proposed WVAT on the geometry of *weight loss landscapes*. We visualize the loss landscapes by plotting the change in loss as the weight moves along the direction of the weight variation $\Delta \boldsymbol{w}$.

$$L(\boldsymbol{w} + \Delta \boldsymbol{w}) = L(\boldsymbol{w} + \gamma \mathcal{N}(0, \ A\sigma_v{}^2))$$

where $\sigma_v$ varies in the range of $[0, 15]\%$, and moving direction is adjusted with $\gamma = -1$ and $\gamma = 1$. We visualize it as $\sigma_v$ drawn from 10 different samplings.

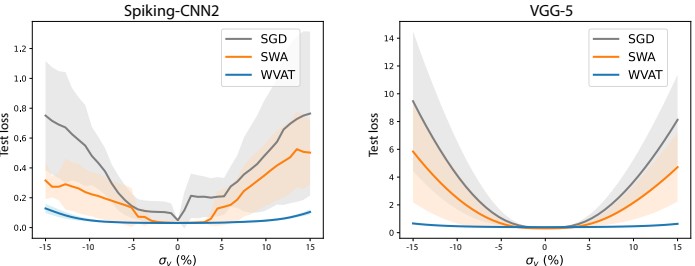

Figure 5: Loss landscapes with respect to model weights. (**left**) Spiking-CNN2 on MNIST, and (**right**) VGG-5 on CIFAR-10.

Figure 5 shows the test loss as a function of the magnitude of the weight variation. Weight loss landscapes become broader and flatter in the order of SGD, SWA, and WVAT, and are uniform for multiple trials. We are taken together with the experimental results in section 4; these experiments show the connection between flat weight loss landscapes and robustness to weight perturbations.

Secondly, we carefully investigate the effect of the proposed WVAT on the weight distribution. As shown in Figure 6 in Appendix A, we verify that WVAT produces larger weights than SGD and SWA. Li et al. (2018) argued that small weights are more sensitive to weight perturbations and make a sharper loss landscape. This claim is consistent with our experimental results. Thus, based on these observations, we demonstrate that WVAT produces larger weights, which makes it less sensitive to weight perturbations and leads to a flatness of the weight loss landscapes.

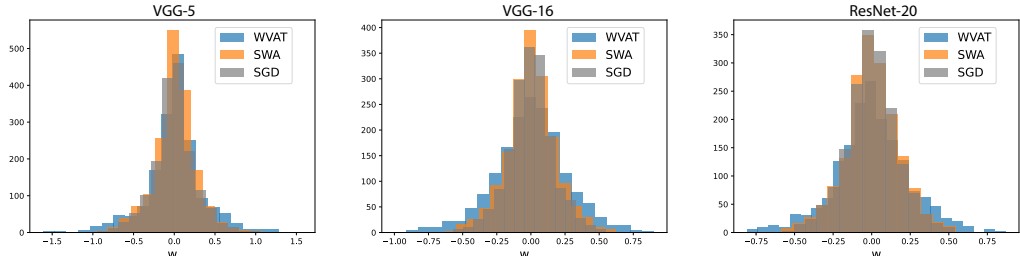

Figure 6: Weight distribution of the first convolutional layer in various network architectures on CIFAR-10.

## 5.2 ROBUSTNESS TO QUANTIZATION

Although WVAT is not for a quantization technique, the fact that WVAT is robust to weight perturbation implies that it is also effective against quantization and noise, another hardware implementation issue. Therefore, we evaluate the robustness of WVAT to quantization and input noise. There are various quantization techniques (Zafrir et al., 2019; van Baalen et al., 2022; Jacob et al., 2018; Kirtas et al., 2022) for small models, but post-training quantization is used to simply investigate the effect of quantization.

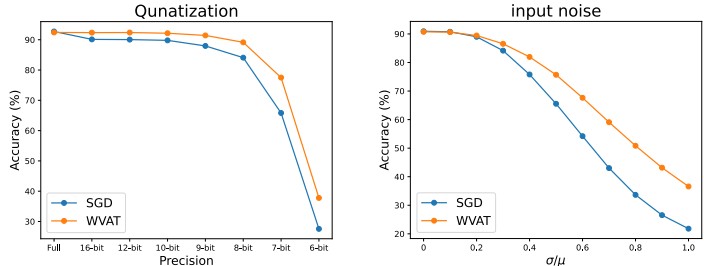

Figure 7: Accuracy of (Left) VGG-16 and (Right) VGG-9 on CIFAR-10.

For quantization and input noise, Figure 7 shows that WVAT provides robustness compared with SGD. We confirm that the proposed method is comprehensively effective for issues for implementing neuro-inspired computing in addition to weight perturbations.

## 6 CONCLUSION

This paper proposes a weight variation-aware training method that is robust to weight perturbations incurred by device variability. For the first time to our knowledge, we investigate and analyze the impacts of weight variations on various benchmark datasets and network architectures. The proposed WVAT effectively minimizes performance degradation by more than 1/10 compared to SGD when there is a weight variation. We propose HSV and GAV to mimic weight variations during the training and present a weight update method that considers the stochastic weight characteristics. We experimentally confirm that WVAT is tolerant to weight perturbations by finding the flat loss landscapes with respect to weight. This method is a hardware-oriented training method at the algorithm level rather than a custom solution at the device level to reduce the performance degradation for the stochastic weight characteristic caused by the inherent variability of the device. Therefore, when the weights trained by WVAT are transferred to hardware, accuracy drop due to device variability can be prevented. It is especially effective in small models.

For limitations and future works, the proposed WVAT computes the gradient-ascent variation, requiring two sequential gradient computations, like AWP and SAM, so there is room for computational improvement. Further research is needed on how to achieve similar performance to WVAT with less computational overhead. We think that studies like Liu et al. (2022) can inspire us to solve this problem.

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

# A APPENDIX

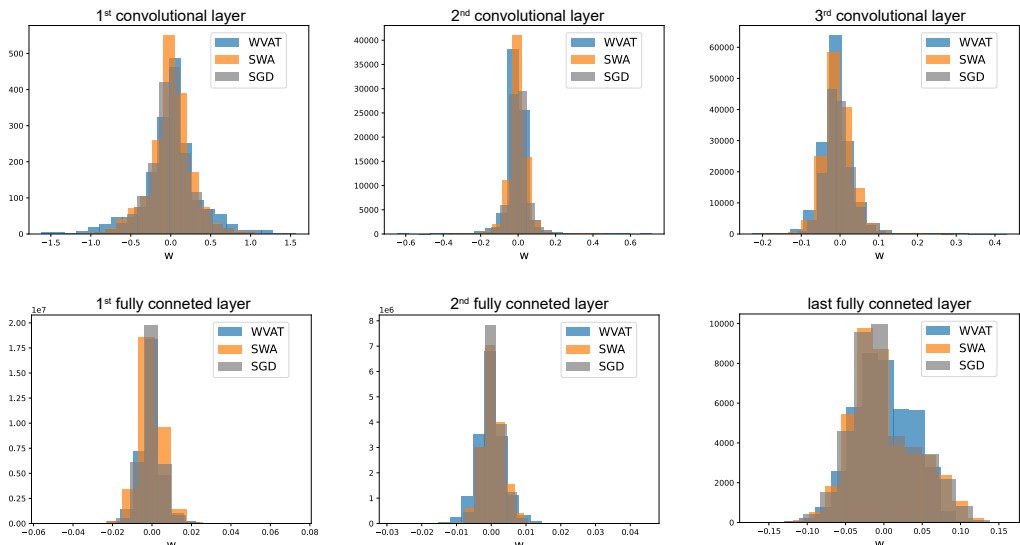

Figure 8: Weight distribution of the entire network layer of VGG-5 on CIFAR-10.

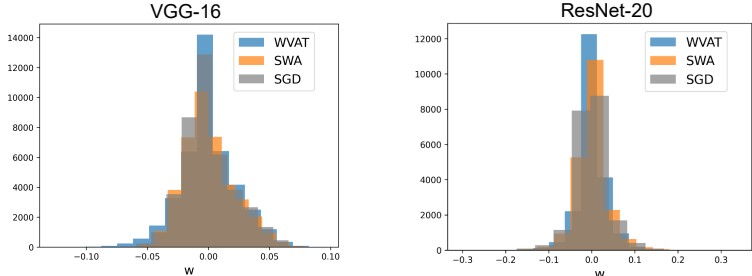

Figure 9: Weight distribution of the last FC layer of VGG-16 and ResNet-20 on CIFAR-10.

