# OpenReview forum: "A Weight Variation-Aware Training Method for Hardware Neuromorphic Chips"
_ICLR.cc/2023/Conference — Submitted to ICLR 2023_

### Official Review · Reviewer_RFsq · 2022-10-21

**Confidence:** 3
**Correctness:** 4
**Technical Novelty And Significance:** 3
**Empirical Novelty And Significance:** 3
**Recommendation:** 6

**Clarity, Quality, Novelty And Reproducibility:**

The paper is mostly well-written and easy to follow. Some editorial improvements could be made (e.g. the sentence "Therefore, the rapid increasing direction is the solution to the problem" -> "Therefore, the most rapidly increasing direction is ..." / A new line with the value 98.09 on page 6).

The paper relates this work to other relevant topics, such as adversarial weight perturbation (AWP) and sharpness-aware minimization (SAM). However, I do wonder if there is a missed opportunity to also relate this to meta learning (see MAML, Reptile, etc.), where each "task" could be the task of inference under perturbed weights (if not, please ignore). The paper presents novelty as evidenced by the experimental improvements over methods like AWP and SAM.

Source code is promised to be released.

To readers outside the neuromorphic literature, it is a bit unclear how close this type of hardware is to productization. I know there are some startups claiming to make chips like this, but I don't know enough to judge what impact this line of work is right now. It would help if the paper discussed how and when this type of algorithm could become used in practice.

**Strength And Weaknesses:**

Strengths
- The training strategy is a combination of weight noise sampling during training (HSV, the more obvious training method) and a minimax loss that assumes perturbation in the worst direction. A combination of both gives the best results and tests on both artificial NNs as well as spiking NNs.
- It is interesting that methods like AWP and SAM do not exhibit better robustness in this regard. This motivates the paper.
- Ultra-low power devices that this is discussing can have significant societal benefits, not the least in the medical domain.

Weaknesses:
- It is never verified on real hardware. It is unknown how this Gaussian perturbation distribution actually corresponds to real hardware. One of the criticisms of prior art was that it was too specific to a particular hardware. That must mean there is a great variation, so does this simple synthetic Gaussian really represent the real world? If the benefit of this is that one training algorithm can work for many types of hardware, shouldn't that also be simulated in the evaluation protocol?

**Summary Of The Paper:**

This paper designs an algorithm for neuromorphic chips, where weights in the final product can look different due to manufacturing imperfections. Assuming the weights are perturbed with a known distribution, this devices an alternative to SGD that aims to minimize the generalization loss with expectation not just over the data distribution, but also the weights distribution. It does this showing improvements over SGD, as well as other methods that would seem to be beneficial for this type of generalization.

**Summary Of The Review:**

This presents a novel algorithm specifically targeting neural network settings where final weights are hard to control exactly due to manufacturing imperfections. This does not affect the vast majority of neural networks running on smart devices today, which does limit the impact, but it could become increasingly important in the future and work in this direction seems motivated. It is sound work with good experimental results. Due to some question marks around potential practical impact and the methods being fairly straightforward, I rate this as a weak accept.

---

> ### Author Response · Authors · 2022-11-19
> **Revision letter**
>
> We thank all the reviewers for their comments. In the following we have attached a point-by- point response to these comments.
>
> First of all, we will release the source code soon for reproducibility.
>
> >It is never verified on real hardware. It is unknown how this Gaussian perturbation distribution actually corresponds to real hardware. One of the criticisms of prior art was that it was too specific to a particular hardware. That must mean there is a great variation, so does this simple synthetic Gaussian really represent the real world? If the benefit of this is that one training algorithm can work for many types of hardware, shouldn't that also be simulated in the evaluation protocol?
> * Many studies have reported that fabricated synaptic devices have Gaussian distribution [1]-[3]; therefore, we use a Gaussian distribution when generating HSV. That is, it is a method for simulating a situation in actual devices. In addition, considering that the standard deviation of the fabricated devices is usually 5% [3], [4], we set that a 10% standard deviation was simulated during training for more stable results. The size of the variance reflected in training can be determined according to the variance of the actual device.
>
> [1] Gong, Nanbo, et al. "Signal and noise extraction from analog memory elements for neuromorphic computing." Nature communications 9.1 (2018): 1-8.
> [2] Boybat, Irem, et al. "Neuromorphic computing with multi-memristive synapses." Nature communications 9.1 (2018): 1-12.
> [3] Yu, Shimeng, et al. "Stochastic learning in oxide binary synaptic device for neuromorphic computing." Frontiers in neuroscience 7 (2013): 186.
> [4] Joshi, Vinay, et al. "Accurate deep neural network inference using computational phase-change memory." Nature communications 11.1 (2020): 1-13.
> [5] Wan, Qingzhou, et al. "Emerging artificial synaptic devices for neuromorphic computing." Advanced Materials Technologies 4.4 (2019): 1900037.

---

> > ### Comment · Reviewer_RFsq · 2022-12-12
> > **Feedback acknowledgement**
> >
> > Thanks for clarifying that the Gaussian assumption is sound and well-documented in the literature.
> >
> > This doesn't fundamentally change my assessment, especially reading the fellow reviews. I will keep my score as is, but I will not disagree strongly if the paper is ultimately rejected.

---

### Official Review · Reviewer_XxLX · 2022-10-25

**Confidence:** 3
**Correctness:** 3
**Technical Novelty And Significance:** 2
**Empirical Novelty And Significance:** 2
**Recommendation:** 5

**Clarity, Quality, Novelty And Reproducibility:**

The paper is confusing, requires multiple passes to understand the scattered details of the implementation
There is some novelty in the joint use of two noise sources. However, the introduction of additional hyperparameters is undesirable.
Sufficient details are provided that should allow to reproduce these experiments.

**Strength And Weaknesses:**

Strengths:
- although the two formulation of the proposed noise is definitely not novel, their threshold-driven combination might be
- experiments compare a range of models across fundamentally different architectures
- experiments do show (Fig. 5) that the proposed technique achieves remarkably flatter losses than the alternatives

Weaknesses:
- paper is often confusing, often lacking in details and clear explanations, it requires further proofreading
- the method introduces two hyperparameters, $w_{th}$ and $\gamma$, for the noise source selection and noise scaling, respectively. It's unclear and never motivated why the selected default values (chosen on the basis of VGG5, CIFAR-10 experiments) would be the appropriate selection across different models
- some level of noise are simulated ($\sigma_v = 5 or 10%$ in Table 1-3) but there is no correlation of this additional parameter with the incidence of hardware noise, which this technique is supposed to address. There is not attempt to establish this relationship at all (not even with a comparison with the literature). The shape of the real hardware noise may also dramatically differ from what is being used here at test time

Other comments and corrections:
- I don't understand the meaning of this sentence in section 1: "we expect that WVAT will help the development of society related to hardware implementation"
- fig 1 shows accuracy drop in the presence of "weight variation". I assume these variations are weight noise added entirely at the software level (simulated device variability). Is this correct? What variations are being simulated in this figure?
- section 3: when $\gamma$ is introduced, it should be specified that it is an additional hyperparameter (this is only mentioned later).
- section 4: SGD-TTV explanation is unclear: "which means when the weight of the trained model perturbs during a test phase". Does this mean the perturbation is only applied at test time? What level of perturbation is being applied?
- figure 4a does not specify what $\gamma$ is being used. figure 4b results do not specify what $w_{th}$ is used and do not appear to match numerically with those in figure 4a (for example, $\gamma=1$ gives $WVAT-TTV \sim 88%$, which in no case matches fig. 4a WVAT-TTV curve).
- section 4.1: I suppose $\alpha$ is the learning rate but it is never defined
- fig. 6 shows a single layer per network. Some visualization of the distribution broadening across the full networks would be more appropriate
- section 5.2: what does it mean that post training quantization "is mainly used as a baseline"?

Other typos:
- fig 4b: "range coefficeint"
- fig 7: "qunatization"


**Summary Of The Paper:**

This paper introduces two sources of noise applied during training to flatten the loss landscape and make weight more resilient to device variability: Hardware-simulated variation (HSA, gaussian noise which scales with the width of the weight distribution of each layer) and Gradient-ascent variation (GAV, added noise in the direction of steepest gradient). The proposed algorithm selects randomly between HSA and GAV during training, using (a hyperparameter $w_{th}$ as threshold and redrawing the selection at every batch. Results are presented on several neural network and spiking neural network models, on basic image classification datasets.

**Summary Of The Review:**

The authors demonstrated that there is value in the idea of alternating sources of noise during training to achieve flatter losses which make the model more robust to some type of noise at inference time. A correlation between actual noise encountered in hardware and this software simulated noise is lacking entirely, so it is hard to draw a conclusion on the actual effectiveness of the proposed technique in real world scenarios. The paper also necessitates extensive proofreading and polishing.

---

> ### Author Response · Authors · 2022-11-19
> **Revision letter 1**
>
> We thank all the reviewers for their comments. In the following we have attached a point-by- point response to these comments.
>
> First of all, we will release the source code soon for reproducibility.
>
> >paper is often confusing, often lacking in details and clear explanations, it requires further proofreading
> *  Typo and unclear parts have been corrected to be more explicit and detailed. Proofreading will be done later.
>
>
> >the method introduces two hyperparameters $w_{th}$,  and $\gamma$, for the noise source selection and noise scaling, respectively. It's unclear and never motivated why the selected default values (chosen on the basis of VGG5, CIFAR-10 experiments) would be the appropriate selection across different models
> * We added experiments on hyperparameters for several models in Figure 4 of the revision paper (VGG-9, ResNet-20 on CIFAR-10 and VGG-16 on CIFAR-100).
> $w_{th}$ is about which variation is selected, and we set empirically default value, which achieves nearly the similar performance as SGD and is robust to weight perturbation. However, since $w_{th}$ may vary depending on the network architectures and datasets, $w_{th}$ for each case is listed in Section 4.
> Regarding $\gamma$, experiments show that it is efficient to reflect the same magnitude of perturbations during test time ($\gamma$ =1).
>
>
> >some level of noise are simulated ( in Table 1-3) but there is no correlation of this additional parameter with the incidence of hardware noise, which this technique is supposed to address. There is not attempt to establish this relationship at all (not even with a comparison with the literature). The shape of the real hardware noise may also dramatically differ from what is being used here at test time
> * Many studies have reported that fabricated synaptic devices have Gaussian distribution [1]-[3]; therefore, we use a Gaussian distribution when generating HSV. That is, it is a method for simulating a situation in an actual device.
> In addition, considering that the standard deviation of the fabricated devices is usually $\sim$5\% [3], [4], we set that a 10\% standard deviation was simulated during training for more stable results.
> The size of the variance reflected in training can be determined according to the variance of the actual device.
>
> [1] Gong, Nanbo, et al. "Signal and noise extraction from analog memory elements for neuromorphic computing." Nature communications 9.1 (2018): 1-8.
> [2] Boybat, Irem, et al. "Neuromorphic computing with multi-memristive synapses." Nature communications 9.1 (2018): 1-12.
> [3] Yu, Shimeng, et al. "Stochastic learning in oxide binary synaptic device for neuromorphic computing." Frontiers in neuroscience 7 (2013): 186.
> [4] Joshi, Vinay, et al. "Accurate deep neural network inference using computational phase-change memory." Nature communications 11.1 (2020): 1-13.
> [5] Wan, Qingzhou, et al. "Emerging artificial synaptic devices for neuromorphic computing." Advanced Materials Technologies 4.4 (2019): 1900037.

---

> > ### Author Response · Authors · 2022-11-19
> > **Revision letter 2**
> >
> >
> >
> > >I don't understand the meaning of this sentence in section 1: "we expect that WVAT will help the development of society related to hardware implementation"
> > *  We aims to close the gap between the ideal(software) and the real (hardware) implementation.
> > Therefore, WVAT is expected to help those suffering from performance drops caused by device variability in actual hardware implementation.
> >
> > >fig 1 shows accuracy drop in the presence of "weight variation". I assume these variations are weight noise added entirely at the software level (simulated device variability). Is this correct? What variations are being simulated in this figure?
> > *   Yes, when the weight of the trained model is transferred to the hardware device, we simulates the situation in which the weight perturbs due to the device variability. The magnitude of the weight variation $\sigma_v$ is 10%.
> >
> >
> > >section 3: when  is introduced $\gamma$, it should be specified that it is an additional hyperparameter (this is only mentioned later).
> > *  We mentioned a $\gamma$ in Section 3, **HSV**.
> >
> > >section 4: SGD-TTV explanation is unclear: "which means when the weight of the trained model perturbs during a test phase". Does this mean the perturbation is only applied at test time? What level of perturbation is being applied?
> > *  TTV refers to the weight perturbation that occurs during test (inference) time when weights trained in software are implanted in hardware. The petrurbation level is $\sigma_v$.
> >
> >
> >
> > >figure 4a does not specify what  is being used. figure 4b results do not specify what  is used and do not appear to match numerically with those in figure 4a (for example,  gives , which in no case matches fig. 4a WVAT-TTV curve).
> > *  Figure 4(b) is for VGG-9, but we wrote this part wrong in the original manuscript.
> > We revised it according to the reviewer's comments.
> >
> >
> > >section 4.1: I suppose $\alpha$  is the learning rate but it is never defined
> > *  We mentioned a step-length parameter $\alpha$ in Section 3, **GAV**.
> > $\alpha$ determines how much variation is given in the direction of the worst case.
> >
> >
> > >fig. 6 shows a single layer per network. Some visualization of the distribution broadening across the full networks would be more appropriate
> > *  We add the weight distribution of the entire network layer of VGG-5 and the last FC layer of VGG-16 and ResNet-20 in the Appendix.
> > It is confirmed that the WVAT tends to widen the weight distribution.
> >
> > >section 5.2: what does it mean that post training quantization "is mainly used as a baseline"?
> > * We referred to as baseline because it is a method basically provided by libraries.
> > https://www.tensorflow.org/lite/performance/post_training_quantization
> > https://pytorch.org/blog/introduction-to-quantization-on-pytorch/#post-training-static-quantization
> > It is also used to compare the proposed methods in the following papers.
> > Kirtas, M., et al. "Quantization-aware training for low precision photonic neural networks." Neural Networks 155 (2022): 561-573.
> > However, we remove this sentense to avoid misunderstanding.

---

### Official Review · Reviewer_oSwx · 2022-10-26

**Confidence:** 3
**Correctness:** 3
**Technical Novelty And Significance:** 2
**Empirical Novelty And Significance:** 2
**Recommendation:** 5

**Clarity, Quality, Novelty And Reproducibility:**

* This paper is clear and with decent quality.

**Strength And Weaknesses:**

Strength:
* The paper is well struct and easy to follow. Results are clearly listed in tables and figures.

Weaknesses:
* I think the claim of "hardware-oriented" is a bit of stretch as the core of the target problem in improving robustness on process variation mostly occurs in neuromorphic computing with analog in particular.
* For having a process variation of 10%, does this mean testing with all weights with 10% shift, or up to 10% shift?
* For figure 7, while I do understand that quantisation is an issue, comparing the performance against SGD rather than quantisation techniques seems a bit misleading. The same applies to input noise, these are covered by network robustness and attack.


**Summary Of The Paper:**

This paper proposed a more robust training method for neurotrophic devices

**Summary Of The Review:**

This paper does propose a new training method to improve robustness towards wright variation, the applicability does not occur very broad to me other than for memristive devices. A few claims and comparison in quantisation and input noise should be fairer.

---

> ### Author Response · Authors · 2022-11-19
> **Revision letter**
>
> We thank all the reviewers for their comments. In the following we have attached a point-by- point response to these comments.
>
> >I think the claim of "hardware-oriented" is a bit of stretch as the core of the target problem in improving robustness on process variation mostly occurs in neuromorphic computing with analog in particular.
> * We agree WVAT is a method for neuromorphic computing with analog sense.
> However, in order to implement ultra-low-power NN execution, it is advantageous to use an analog synaptic device that can express weight values with a single synaptic device.
> It is true that memristive devices are mainly used as analog synaptic devices, but considering compactness and energy efficiency, which are the critical requirements of edge devices, We think this is a worthy direction and will become more important in the future.
>
> >For having a process variation of 10%, does this mean testing with all weights with 10% shift, or up to 10% shift?
> * $\sigma_v$ of 10% means that values sampled from a Gaussian distribution with standard deviations of 10% are added.
>
> >For figure 7, while I do understand that quantisation is an issue, comparing the performance against SGD rather than quantisation techniques seems a bit misleading. The same applies to input noise, these are covered by network robustness and attack.
> * Similar to "robustness to label noise" experiment in the SAM paper, it is an experiment to measure the effects of WVAT in quantization.
> For that reason, the comparison is made with SGD, which is not a quantization technique.
> In other words, WVAT is not a quantization technique, but it can be tolerant with quantization in that changes occur in weights trained by software.
> However, as the reviewer commented, there may be misunderstandings, so this part was clearly mentioned and will add a comparison experiment with quantization techniques.

---

### Official Review · Reviewer_giaL · 2022-10-30

**Confidence:** 3
**Correctness:** 3
**Technical Novelty And Significance:** 2
**Empirical Novelty And Significance:** 3
**Recommendation:** 3

**Clarity, Quality, Novelty And Reproducibility:**

The paper is generally well written. However, the paper would benefit from more details in Section 3. The paper only spend little portion of the paper to explain the main idea.

The paper makes reasonable contribution in terms of considering the variability through imitation (HSV) and considering the worst cases of the variation through (GAV).

Details in Section 3 seems to be rather weak, which may limit reproducibility.

**Strength And Weaknesses:**

+ SNNs are interesting research topic with huge potential for ultra-low-power NN execution.
+ Closing the gap between the ideal vs real execution (with perturbation) is important.

- More details on Section 3 would help.

**Summary Of The Paper:**

Spiking Neural Networks mimic the biological nervous systems for ultra-low-power NN execution.
However, it can be sensitive to the weight perturbations that may lead to significant performance drop.
The paper aims to understand the causal relationship between the perturbations and the performance drop to devise a Weight Variation Aware Training Method.
The paper provides experimentation on small networks to show that it can reduce accuracy degradation.

**Summary Of The Review:**

The paper aims to close the gap between the ideal and the real implementation of the SNNs by devising a variation-aware training. Considering the potential upsides of the SNNs such as the low power NN execution, the paper works on an important topic. While some improvements can be made, the paper also does a good job in explaining the problem and the key ideas.

The paper provides two ideas Hardware-simulated variation (HSV) which acts as a imitation model of the hardware variability. Similar idea is presented in the following paper which should be noted. However, considering the difference (DNN vs SNN) this seems like a neat contribution.
* Ghodrati, Soroush, et al. "Mixed-Signal Charge-Domain Acceleration of Deep Neural Networks through Interleaved Bit-Partitioned Arithmetic." Proceedings of the ACM International Conference on Parallel Architectures and Compilation Techniques. 2020.
In addition to HSV, the paper develops Gradient-ascent variation which considers the worst case.

The paper runs experiments to show the effectiveness. It seems to make a thorough comparison against baselines (SGD and SWA). The paper also provides some discussions. While the experimentation on SNNs seem reasonable, experiments with ANNs seem rather weak. Adding some datapoints (larger networks, datasets) would make the claims stronger in the paper.

* Typo in Figure 7: Qunatization -> Quantization

---

> ### Author Response · Authors · 2022-11-19
> **Revision letter**
>
> We thank all the reviewers for their comments. In the following we have attached a point-by- point response to these comments.
>
>
> >More details on Section 3 would help.
> * Section 3 has been supplemented by reflecting the reviewer's comments in revised manuscript.
> The source code will be released soon for reproducibility.
>
> >While the experimentation on SNNs seem reasonable, experiments with ANNs seem rather weak. Adding some datapoints (larger networks, datasets) would make the claims stronger in the paper.
> * The results of experiments on a larger network on CIFAR-100 are as follows.
> |Model|Method|Acc (%)|Acc (%) $\sigma_v$=10%|
> |:------:|:---:|:---:|:---:|
> |ResNet-164|SGD|77.9|36.933|
> |------|SWA|79.43|54.38|
> |------|WVAT|74.52|58.96|
> |WideResNet 28x10|SGD|80.57|69.23|
> |------|SWA|82.31|73.29|
> |------|WVAT|81.26|76.53|
>
>
> This is an initial experimental result, and we believe that better results can be obtained through hyperparameter settings, and we will reflect the final experimental results later. Experiments on larger dataset sets will also be added.

---

### Decision · Program_Chairs · 2023-01-20

**Decision:**

Reject

**Justification For Why Not Higher Score:**

Reviewer scores are rather low [3,5,6,5], with even the marginally positive reviewer wrote that they "will not disagree strongly if the paper is ultimately rejected.". Moreover, this paper has several serious issues, as mentioned in the summary.

**Justification For Why Not Lower Score:**

N/A

**Metareview: Summary, Strengths And Weaknesses:**

This paper suggests a method for making neural networks (both standard and spiking) robust to parameter variation. This is an important problem, but this paper has many weaknesses, for example:
1) The method is compared only against only very basic baselines (SGD, SWA). Why not compare against previous methods, e.g. the analog-aware training from the Joshi et al. reference? Also, why not compare against SAM? Given the background, it is strange that, in Figure 1, the results of SAM are not shown, though it is rather different from AWP, and much more related to the current method.
2) The datasets are small. For example, Joshi et al. have experiments on ImageNet.
3) The paper clarity and quality of writing should be improved as the reviewers noted, and the equations have issues (see below some additional examples).
4) The novelty of the method seems limited, as HSV is similar to Joshi et al. and GAV seems similar to SAM (with p=q=2). It is hard to know for sure how similar, because of the clarity issues mentioned above.


**Some issues with equations:
1) In section 3, I do not understand the relation between \delta w, v(w) and w_v. Are they all the same? If not, how are they related? Also, what determines the norm of v(w) in the weight update with GAV?
2) In the equations before GAV, it should be I instead of 1, right? In any case, it seems like a trivial equation.
3) In the AWP equation, v appears on both sides of the equation. I think it should be '\leftarrow' instead of '='.

***minor:
Joshi et al. it has a duplicated citation.